# The Efficacy of Platelet-Rich Plasma Injection Therapy in Obese versus Non-Obese Patients with Knee Osteoarthritis: A Comparative Study

**DOI:** 10.3390/jcm13092590

**Published:** 2024-04-28

**Authors:** Juho Aleksi Annaniemi, Jüri Pere, Salvatore Giordano

**Affiliations:** 1Department of Surgery, Welfare District of Forssa, 30100 Forssa, Finland; jualan@utu.fi (J.A.A.); juri.pere@fshky.fi (J.P.); 2Department of Surgery, Satasairaala Hospital, Satakunta Wellbeing Services County, 28500 Pori, Finland; 3Department of Plastic and General Surgery, Turku University Hospital, The University of Turku, 20500 Turku, Finland

**Keywords:** platelet-rich plasma, injection therapy, knee osteoarthritis, PRP, obesity, WOMAC

## Abstract

**Background/Objectives**: Obesity is a common comorbidity in knee osteoarthritis (KOA) patients. Platelet-Rich Plasma (PRP) injection therapy may mitigate KOA. To further clarify potential patient selection for PRP injection therapy, we compared the outcomes in patients with different body mass index (BMI). **Methods**: A total of 91 patients with mild to moderate KOA were treated with three intra-articular PRP injections at 10 to 14-day intervals. Range of motion (ROM), Western Ontario and McMaster Universities Osteoarthritis Index (WOMAC), and Visual Analogue Scale (VAS) were documented before and after the injections at 15 days, 6 months, 12 months, and at the last follow-up. Outcomes were compared between patients with a BMI over 30 kg/m^2^ (obese, n = 34) and under 30 kg/m^2^ (non-obese, *n* = 57). **Results**: Significant difference during the follow-up was detected in WOMAC score at the last follow-up favouring BMI under 30 group [17.8 ± 18.8 versus 10.5 ± 11.7, *p* = 0.023]. The odds ratio (OR) in BMI over 30 kg/m^2^ group for total knee arthroplasty was 3.5 (95% CI 0.3–40.1, *p* = 0.553), and OR for any arthroplasty was 7.5 (95% CI 0.8–69.8, *p* = 0.085) compared to non-obese patients. **Conclusions**: Obese patients benefitted from PRP injections in KOA but there is a minimal difference favouring non-obese patients in symptom alleviation in follow-up stages after 12 months. The risk of arthroplasty is higher for obese KOA patients.

## 1. Introduction

The incidence of obesity and overweight is continuously increasing, with estimates of 38% of the population being overweight and up to 20% being obese by the year 2030 [1]. The definition of overweight is body mass index (BMI) greater than ≥25 kg/m^2^ to 29 kg/m^2^, while obesity is defined as BMI greater than 30 kg/m^2^ [1]. Obesity is associated with metabolic abnormalities which have negative effects considering general health and these abnormalities may advance osteoarthritis (OA) on a molecular level [2]. Function and clinical outcomes and consequences of knee osteoarthritis (KOA) are directly associated with the level of obesity with worsening results as the BMI rises [3].

Arthroplasty for KOA in obese and morbidly obese patients with BMI ≥ 40 kg/m^2^ is heavily debated due to the increased rate of complications [4]. The complication rate is tied closely to the increase in BMI, with studies indicating that, for every 15 patients denied surgery due to BMI ≥ 40 kg/m^2^, one is saved from a major complication, and similarly for BMI ≥ 35 kg/m^2^ and ≥30 kg/m^2^ the ratios are 1/17 and 1/18 patients, respectively [4,5]. It is well documented that obese and morbidly obese patients have a significantly higher complication rate in arthroplasty surgery than non-obese patients [6]. Therefore, alternative options to treat these patients throughout the progression of OA are necessary in order to avoid or delay arthroplasty.

Despite the controversy surrounding PRP treatments, they are shown to reduce clinical symptoms of KOA for roughly 12 to 24 months per treatment [7,8]. However, there is little information about their efficacy in obese patients and patient selection altogether. Small but statistically significant differences were detected favouring PRP over HA in terms of patient satisfaction, overall experience of symptom-free time after the treatment, rate of re-intervention, and clinical symptom scores [7,8]. Overweight or obese patients seem to have similar results in KOA symptom scores at the first 2 months, but after that at 6 and 12 months, PRP patients experienced better functional improvement and pain relief than hyaluronic acid (HA) patients [9].

The impact of obesity on PRP treatment efficacy has been evaluated through a previous meta-analysis [9]. Luo et al. (2020) found that intra-articular injection of PRP had better long-term outcomes in pain and functional relief than hyaluronic acid for overweight or obese patients with knee OA [9]. Recent findings from placebo-controlled studies strongly indicate that platelet-rich plasma (PRP) emerges as a promising treatment modality for knee osteoarthritis (KOA). These studies demonstrate that PRP not only effectively reduces symptoms over a prolonged period, surpassing the efficacy of a placebo for at least 24 months, but also yields substantial benefits in terms of significantly attenuating tibiofemoral cartilage loss. Notably, MRI follow-ups conducted over a span of 5 years consistently reveal marked reductions in tibiofemoral cartilage degeneration following PRP treatment, underscoring its potential as a durable therapeutic intervention for KOA [10]. Nevertheless, patient selection for PRP treatments is also heterogenic and it is unknown whether there are patient groups that would benefit more or in turn less from the treatment than others. Therefore, the aim of the present study was to determine the efficacy of autologous intra-articular injections of PRP in KOA in obese versus non-obese patients to clarify and further improve the patient selection for the use of PRP treatments. Usually, obese patients are excluded from studies about intra-articular PRP injections [11,12]. We hypothesized that obese patients might have less benefits from PRP treatments in delaying arthroplasty.

## 2. Materials and Methods

This is a retrospective cohort study including a total of 91 patients with symptomatic KOA. Patients had mild to moderate KOA, ranging from Kellgren–Lawrence (KL) grade 1 to 3. Patients in this study were consecutive and had received PRP injections to their knees between January 2014 and October 2017 at the Welfare District of Forssa, Finland. Ethical principles of the World Medical Association Declaration of Helsinki were followed, and the Institutional Review Board approved the study. Individual informed consent was waived due to the retrospective nature of the source data and their de-identification.

The patients’ demographic data were collected meticulously from the electronic medical records including preintervention and follow-up outpatient clinics. Inclusion criteria were having data on the Visual Analogue Scale (VAS) of 30 to 100 before treatment, mild to moderate KOA in radiographs (KL grade 1 to 3), and age over 18 and below 90 years. The exclusion criteria comprehended major systemic diseases or infections (such as active fulminant rheumatoid disease, immunodeficiency, haematological disease), clinically relevant hip OA of the same side, pregnancy or possibility of pregnancy, and age below 18 or above 90. This study’s population mirrors a characteristic demographic found in Finland, comprising individuals afflicted with symptomatic mild to moderate knee osteoarthritis (KOA). This cohort encapsulates the diverse spectrum of patients commonly encountered in public healthcare settings within Finland, thereby possibly enhancing the generalizability and applicability of the study’s findings to real-world scenarios.

Patients were followed up with a physical examination, comprehensive of Western Ontario and McMaster Universities Osteoarthritis Index (WOMAC) questionnaire, VAS, and range of motion (ROM) of the knee. Follow-up points were considered at 15 days, at 6 months, at 12 months, and at the last follow-up after injections.

For the purpose of this study, patients were divided into two groups on the basis of BMI. Group A had BMI ≥ 30 kg/m^2^ and group B had BMI ≤ 30 kg/m^2^. Group A consisted of 34 patients and group B had 57 patients. All the patients received three injections of autologous PRP at approximately two-week intervals. The PRP was manufactured with the commercial Glo PRP kit (GloFinn corporation, Salo, Finland). An experienced and trained nurse drew the patients’ blood up to 30 mL, which was then centrifuged twice. Sodium citrate was used as an anticoagulant to prevent platelet aggregation and clotting. First centrifugation lasted 5 min at 1200 rpm, then excess red blood cells were removed, and the product was centrifuged for a second time 10 min at 1200 rpm. Platelet concentration in the final product was 4 to 8 times above physiological normal values, and the white blood cell concentration was doubled according to the manufacturer. The PRP obtained for this study exhibited an elevated white blood cell count surpassing baseline levels, thus, categorizing it as leukocyte-rich PRP as per previously established definitions. However, owing to the retrospective design of the present study, comprehensive data or analyses pertaining to the exact platelet-to-white blood cell ratios within the final PRP product were unattainable. The volume of PRP administered into the joint space ranged between 4 to 5 mL, representing a standard protocol for treatment delivery in the study cohort. The intra-articular injections were performed by an experienced orthopedist using aspiration and anatomical landmarks to inject the PRP inside the intra-articular space.

The primary outcome measures were pain and functional outcomes measured as VAS, WOMAC, and ROM of the knee. Secondary outcome measures included the occurrence of any knee arthroplasty and adverse events at follow-up after intra-articular injections.

The indication for PRP intra-articular injections included symptoms and pain due to arthrosis in patients with KL grade 1 to 3 knee OA in radiographic imaging, pre-intervention pain Visual Analogue Scale (VAS) of 30 to 100, and age over 18 years.

The indication for surgical intervention was at the surgeon’s discretion when failure of prolonged nonoperative treatment (in this study physical therapy and pain medication) persisted over 6 months.

Statistical analysis was conducted using SPSS statistical software (IBM SPSS Statistics, version 28, Armonk, NY, USA). Continuous variables were described as mean ± standard deviation. Normality assumptions were established by histograms, Kurtosis, Skewness, and occasionally with Kolmogorov/Smirnov tests. Pearson’s chi-square test, Fisher’s exact test, and *t*-test were employed for carrying univariate analysis, as appropriate, for comparisons between the two study groups according to the BMI (obese, >30 kg/m^2^ versus non-obese, <30 kg/m^2^). A two-sided *p*-value less than 0.05 was considered statistically significant. The post hoc statistical power was calculated to be 47.5% for the primary outcome measure, considering an observed effect size of 0.436 (Cohen’s d).

## 3. Results

The mean follow-up was over 13 months and similar in both groups [Group A 13.6 ± 4.7 months versus Group B 14.9 ± 5.6 months, *p* = 0.254] (Table 1). The preintervention demographic data showed no statistically significant differences. Osteoarthritis grade and symptom scores were similar in both groups (Table 1). Throughout the follow-up, the symptom scores diminished in both groups, but a statistically significant difference was detected in WOMAC only at the last follow-up favoring Group B, the non-obese patients [Group A 17.8 ± 18.8 versus Group B 10.5 ± 11.7, *p* = 0.023] (Table 2, Figure 1). Furthermore, Group A had more arthroplasties than Group B, but this difference remained statistically insignificant. The odds ratio (OR) for total knee arthroplasty was 3.5 (95% CI 0.3–40.1, *p* = 0.553), and the OR for any arthroplasty was 7.5 (95% CI 0.8–69.8, *p* = 0.085). Group B experienced slightly more adverse events from the injections than Group A, but this finding was not statistically different (Table 2).

## 4. Discussion

This is a study attempting to find possible differences in BMI in PRP injection outcomes for KOA.

Both the obese and non-obese groups benefitted from the PRP treatments for their KOA, but patients with BMI under 30 kg/m^2^ seem to have a slight edge over patients with BMI over 30 kg/m^2^ in an over one-year follow-up. The differences in clinical symptom scores and number of arthroplasties are minimal in terms of clinical practice but do hint that obese patients with KOA are at greater risk of arthroplasty than non-obese patients. Only the WOMAC score revealed a statistically significant difference between the two groups favoring Group B over Group A at the last follow-up point. VAS and ROM remained indifferent throughout the follow-up. The gradual return of symptoms occurred earlier in group A with a higher BMI than in group B.

Differences in WOMAC at the last follow-up may be due to Group A having a higher BMI, therefore, having more mechanical stress and a greater tendency for smouldering inflammation in the knee joint, and thus, provoking the symptoms easier than Group B with a lower BMI. Previous studies showed that the progression of OA is driven by a complex interplay of mechanical stress, several matrix metalloproteinases, and cytokines, that together cause inflammation and apoptosis, leading to the degeneration and eventual destruction of the cartilage [13,14]. The growth factor composition of PRP is also similar in obese and non-obese patients, therefore the treatment is likely to have similar effects despite BMI differences [15]. The groups did not have any significant differences in their VAS, ROM, WOMAC, or grading of their KOA before the treatments. Moreover, there were no differences in the other demographics of the two groups, other than mean BMI. Therefore, it is probably because obesity itself implies an earlier return of the symptoms.

Interestingly, both ROM and VAS remained insignificant between the groups throughout the follow-up, although there was a higher spread in mean VAS in Group A than in Group B. Because of the statistically non-significant difference in VAS, it may be that pain alone may not explain the difference in WOMAC score. Further investigation into this matter may be warranted.

The OR for total knee arthroplasty was 3.5 (95% CI 0.3–40.1, *p* = 0.553) while for any arthroplasty OR was 7.5 (95% CI 0.8–69.8, *p* = 0.085). This difference between any arthroplasty tended to be near significant. To clarify this, the study population should have been larger and the follow-up time much longer. When considering the significance of this in clinical practice, obese patients end up in arthroplasty surgery with greater risk and therefore are also at greater risk of devastating complications. Given the mean follow-up of over a year, perhaps obese patients would have time to address their own lifestyle choices and weight control by undergoing PRP treatments before arthroplasty to reduce the potential risk for complications. On the other hand, the non-obese patient seems to avoid arthroplasty after over a year’s follow-up and has minimal symptoms in the knee compared to the preintervention symptom scores.

Both groups experienced adverse events from the injections and a total of four adverse events were documented, one in Group A and three in Group B (Table 2). All the adverse events in both groups were prolonged post-injection pain in the knee with mild effusion of the knee joint and none of them were considered serious adverse events. The adverse events are probably inherent to injection therapy in general and require no additional treatments as the symptoms resolved spontaneously in 3 to 5 days.

Non-obese patients benefit more from the PRP injections than obese patients, however obese patients receive meaningful alleviation to their symptoms but are still at greater risk of arthroplasty than non-obese patients. This may be a piece of useful information in clinical practice when considering different treatment options for different patients with symptomatic KOA. PRP seems to provide long-lasting improvement for the patients and may give time for the patient and the physician to take measures to reduce the risks of possible future arthroplasty due to obesity. PRP injection is a viable and safe alternative option to treat symptomatic KOA, alongside anti-inflammatory drugs (NSAID) and acetaminophen (APAP). Patient preference is crucial in selecting the treatment and some younger patients may prefer injections with long efficacy over oral medication, and in turn elderly patients may have medical conditions or drugs that may not allow them to use APAP or NSAID. This study may contribute to the decision of treatment modality when treating KOA patients of different BMI that are not yet ripe for arthroplasty.

The novelty of this study was identifying differences between obese and non-obese patients with mild to moderate KOA and differentiating the responses to autologous intra-articular knee PRP injections through direct comparison. Our results indicate that PRP seems to be an adequate treatment in both obese and non-obese patients with KL 1 to 3 KOA. Non-obese patients have a slight edge over obese patients in over a year follow-up in WOMAC symptom scores and have a much lesser risk for arthroplasty than obese patients. Earlier return of symptoms is probably due to higher BMI, further strengthening the hypothesis that obesity is one of the key factors attributing to KOA symptoms. Both groups had very few and mild adverse events, which are probably similar in all intra-articular injections. Only a few patients had arthroplasty and even the KL 3 patients received alleviation to their symptoms for at least 12 months. A recent study conducted by Saraf et al. has revealed that demographic factors such as age, gender, or BMI exert no discernible impact on the clinical outcomes observed following PRP injections in patients diagnosed with KOA of K-L grade 2 to 3. This finding underscores the robustness and potential efficacy of PRP therapy across diverse patient profiles within this specific KOA subgroup [16].

The strengths of this study included a meticulous collection of preintervention data including symptom scores, moderate follow-up length, and adequate sample size in both groups representing well all stages of mild to moderate KOA. Arthroplasties and adverse events regarding the injections were carefully documented. The inclusion and exclusion criteria were strict.

The limitations of this study include the retrospective setting, smaller number of patients in group A, and sex ratio leaning towards females over males. We only included patients with KL grade 1 to 3 KOA, because previous studies show that PRP has little or no effect in KL grade four osteoarthritis, therefore, comparing the end-stage KOA is unnecessary [17,18]. There were only a few KL grade 1 patients in both groups, but finding these patients is difficult due to KOA of that grading rarely causing enough symptoms for the patients to actively seek medical help. The PRP used in this study was prepared according to Glo PRP kit instructions (GloFinn corporation, Salo, Finland), which may differ from other studies that used different PRP kits. Given the retrospective design of this study, detailed data or analyses regarding the final platelet or white blood cell counts are regrettably unavailable. Nonetheless, it is important to note that the exclusion criteria encompassed also patients with any haematological conditions, thereby mitigating potential variations in platelet and white blood cell counts, other than those arising from normal physiological fluctuations. This meticulous exclusion process attempted to ensure a more homogenous study population, enhancing the reliability and validity of our findings concerning the efficacy of PRP therapy in KOA management.

## 5. Conclusions

Obese patients receive some help and benefit from the PRP injections as non-obese in short follow-up, but the effects of PRP diminish quicker over time in obese patients, exposing them to a higher risk for arthroplasty than non-obese patients. In summary, the observed differences were minimal, indicating that obese patients experience outcomes comparable to those of non-obese individuals following autologous intra-articular leukocyte-rich PRP injections for mild to moderate symptomatic KOA. These findings underscore the potential efficacy of PRP therapy across diverse patient populations, irrespective of weight status. Identifying a patient who is at higher risk of arthroplasty may give the patient and physician time to make necessary interventions (weight loss, lifestyle) to reduce the surgical risk of arthroplasty or complication risks involved in the arthroplasty of an obese patient.

## Figures and Tables

**Figure 1 jcm-13-02590-f001:**
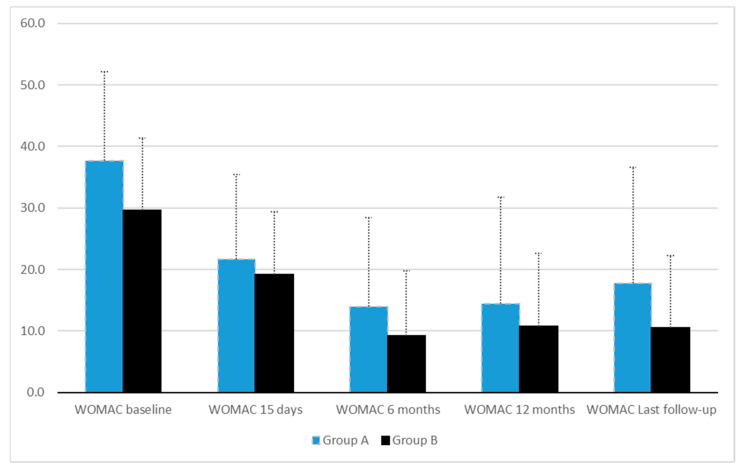
The mean values of Western Ontario and McMaster Universities Osteoarthritis Index (WOMAC) during the follow-up with ±1 standard deviation.

**Table 1 jcm-13-02590-t001:** Baseline characteristics of patients who underwent intra-articular injections of platelet-rich plasma divided according to the body mass index, obese versus non-obese.

	Group A (BMI ≥30 kg/m^2^)	Group B (BMI <30 kg/m^2^)	*p*-Value
	n = 34	n = 57	
Mean age (years)	56.1 ± 9.8	58.3 ± 10.6	0.343
Females	23 (67.6%)	33 (57.9%)	0.382
Mean BMI † (kg/m^2^)	33.6 ± 3.9	26.0 ± 1.9	<0.001
Comorbidity	16 (47.1%)	16 (28.1%)	0.075
Diabetes	4 (11.8%)	3 (5.3%)	0.418
Hypertension	17 (50.0%)	18 (31.6%)	0.118
Cardiac disease	4 (11.8%)	6 (10.5%)	1.000
Smokers	7 (20.6%)	8 (14.0%)	0.560
Osteoarthritis Grade (Kellgren–Lawrence, mean ± SD †	2.35 ± 0.6	2.21 ± 0.6	0.299
Osteoarthritis Grade (Kellgren–Lawrence) I	2 (5.9%)	7 (12.3%)	0.475
Osteoarthritis Grade (Kellgren–Lawrence) II	17 (50.0%)	30 (52.6%)	0.831
Osteoarthritis Grade (Kellgren–Lawrence) III	15 (44.1%)	20 (35.1%)	0.504
Bilateral	4 (11.8%)	9 (15.8%)	0.760
Flexion degree	130.0 ± 20.7	131.2 ± 17.9	0.769
Extension degree	88.5 ± 3.4	88.6 ± 2.9	0.921
VAS pain score (0–100)	64.8 ± 16.3	65.4 ± 18.4	0.882
WOMAC overall	34.7 ± 14.5	29.6 ± 11.8	0.085
Follow-up (months)	13.6 ± 4.7	14.9 ± 5.6	0.254

† BMI; body mass index. SD; standard deviation.

**Table 2 jcm-13-02590-t002:** Outcomes of patients who underwent intra-articular injections of platelet-rich plasma divided according to the body mass index, obese versus non-obese.

	Group A (BMI ≥ 30 kg/m^2^)	Group B (BMI < 30 kg/m^2^)	*p*-Value
	n = 34	n = 57	
Adverse events	1 (2.9%)	3 (5.2%)	1.000
Number of injections	3.1 ± 0.7	3.3 ± 1.4	0.653
Flexion degree (15 days)	132.8 ± 17.9	135.2 ± 15.1	0.499
Extension degree (15 days)	89.0 ± 2.7	89.2 ± 2.1	0.634
VAS pain score (0–100)-(15 days)	36.5 ± 22.9	40.3 ± 19.7	0.395
WOMAC overall (15 days)	21.7 ± 13.7	19.2 ± 10.2	0.744
Flexion degree (6 months)	136.7 ± 15.8	138.3 ± 15.4	0.626
Extension degree (6 months)	89.7 ± 1.2	90.1 ± 4.4	0.618
VAS pain score (0–100)-(6 months)	21.4 ± 24.1	18.2 ± 23.3	0.540
WOMAC overall (6 months)	13.9 ± 14.5	9.2 ± 10.6	0.083
Flexion degree (12 months)	137.0 ± 16.7	138.2 ± 13.6	0.748
Extension degree (12 months)	89.1 ± 2.4	89.4 ± 1.9	0.581
VAS pain score (0–100)-(12 months)	21.3 ± 26.7	21.2 ± 24.3	0.989
WOMAC overall (12 months)	14.4 ± 17.3	10.7 ± 11.9	0.283
Flexion degree (last follow-up)	136.2 ± 16.6	138.7 ± 14.4	0.450
Extension degree (last follow-up)	88.8 ± 2.8	89.4 ± 1.9	0.255
VAS pain score (0–100)-(last follow-up)	27.1 ± 30.0	19.0 ± 21.6	0.142
WOMAC overall last follow-up)	17.8 ± 18.8	10.5 ± 11.7	0.023
Any knee arthroplasty	4 (11.8%)	1 (1.8%)	0.063
Unicompartmental knee arthroplasty	2 (5.9%)	0	0.137
Total knee arthroplasty	2 (5.9%)	1 (1.8%)	0.553

BMI; body mass index. SD; standard deviation.

## Data Availability

The data presented in this study are available on request from the corresponding author.

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
