# Peer review of "The Efficacy of Platelet-Rich Plasma Injection Therapy in Obese versus Non-Obese Patients with Knee Osteoarthritis: A Comparative Study"

_jcm, 2024, doi:10.3390/jcm13092590_

Round 1
Reviewer 1 Report
Comments and Suggestions for Authors
I commend all the authors for their outstanding work and the interesting results presented in this study. This study holds potential for significant insights. However, some minor revisions could enhance the final version of the paper.
1. The Introduction is well-structured and comprehensible.
2. The "Methods" section is clearly described and appears reproducible.
3. It's crucial to provide information about the overall volume of PRP injection post-centrifugation. The noted amount of 10ml of blood may not be sufficient for PRP processing for knee joint applications. An explanation of this limitation would be valuable.
4. Additionally, it would be beneficial to include data on the precise ratio of platelets and leukocytes in accordance with the Ehrenfest classification.
5. Both the "Results" and "Discussion" sections are well-organized and easy to follow.
6. It would be enlightening if the authors could engage more with other studies related to the role of patient characteristics (such as BMI) and the efficacy of regenerative injections.
Once again, I wish you all the best in the further publishing process.

Author Response
Reviewer 1
- The Introduction is well-structured and comprehensible.
Thank you for your comments; this was our aim during the writing process.
- The "Methods" section is clearly described and appears reproducible.
Once again, thank you for your comments. As described in the methods section, the study should be easily reproducible.
- It's crucial to provide information about the overall volume of PRP injection post-centrifugation. The noted amount of 10ml of blood may not be sufficient for PRP processing for knee joint applications. An explanation of this limitation would be valuable.
Thank you for your keen observation. It is important to include the final amount of injected PRP, which was 4-5 mL. This was achieved with three blood draws, comprising approximately 30 mL of venous blood, sufficient to produce 4 – 5 mL of PRP. The method of PRP extraction was described at a general level according to GloFinn Corporation’s kit instructions. Sodium citrate was used as an anticoagulant during the process to prevent the blood from clotting. To further clarify the methodology, we have added this information to the methods section.
- Additionally, it would be beneficial to include data on the precise ratio of platelets and leukocytes in accordance with the Ehrenfest classification.
Thank you for your excellent comment. The manufacturer reports that the platelet concentration is at least four times above physiological levels, and the white blood cell ratio doubles compared to whole blood after centrifugation. However, no analyses of the final product could be made since this is a retrospective study, and patients had already received the injections. Therefore, we do not have data concerning the precise ratios of platelets and leukocytes.
- Both the "Results" and "Discussion" sections are well-organized and easy to follow.
Thank you for your comment; we have endeavored to maintain it in that manner.
- It would be enlightening if the authors could engage more with other studies related to the role of patient characteristics (such as BMI) and the efficacy of regenerative injections.
Thank you for your encouraging comment. We conducted a literature search for studies on autologous intra-articular PRP injection in obese patients with knee osteoarthritis. Although such studies are scarce, we conducted another search and identified one new significant study on the topic, which we have included in the discussion section. Additionally, we incorporated information about the results of placebo-controlled studies on PRP and knee osteoarthritis to clarify the efficacy of regenerative injections in this context. The references list has been reorganized to accommodate the new references and align with the journal's specifications accordingly.
Reviewer 2 Report
Comments and Suggestions for Authors
In the paper of Annaniemi and colleagues authors indicate that obese patients benefitted from PRP injections in knee osteoarthritis. Moreover authors claimed that the risk of arthroplasty is higher for obese knee osteoarthritis patients. Authors came to their conclusions by performing their retrospective cohort study which put some possibility of clinical use of the presented results.
However I have to raise some following comments:
1. Authors performed their study on a relatively small group of patients. Why is the reason of smuch number of individuals? Does this not affect the translation of the observed phenomenon in a larger population?
2. Authors should use appropriate superscript in the formulation kg/m2.
3. Did authors include apyrase or prostaglanding E2 do whole blood before centrifugation to prevent platelets agreggation during PRP obtaining?
4. Why authors did not prepare PRP without contamination of WBC? If their PRP contained WBC, they should not named it as PRP.
5. Did authors verify the platelet count and leukocyte count in obtained PRP? This might be important considering that authors claimed that: "The final PRP contained white blood cells making it leukocyte-rich PRP". What is the percentage of platelets and leukocytes in PRP? There might be some inter-individual differences in count of platelets and leukocytes that could affect the preparation of injected smaples of PRP. If authors measured platelet and leukocyte count, they should inlcude these results in paper.
6. Presentation of results on the Figure 1 is unacceptable. Authos should use a more professional graphical way of presentation their results.
7. Considering the large standard deviation range it is surprising that authors show any significant differences on the Figure 1.
Author Response
Reviewer 2
- Authors performed their study on a relatively small group of patients. Why is the reason of smuch number of individuals? Does this not affect the translation of the observed phenomenon in a larger population?
Thank you for your comment. This study was retrospective and encompassed all patients with symptomatic knee osteoarthritis who had received intra-articular PRP injections, met the inclusion criteria, and did not meet any exclusion criteria. We searched the electronic medical records of the Welfare District of Forssa for patients with mild to moderate knee osteoarthritis who had undergone PRP injection and applied the inclusion and exclusion criteria accordingly, resulting in the identification of 91 patients. Given the retrospective nature of the study, we implemented strict inclusion and exclusion criteria to mitigate confounding factors and ensure that only relevant patients were included. The study population represents a typical Finnish population with symptomatic knee osteoarthritis. To elucidate this point, we have added a comment on the population in the materials and methods section.
- Authors should use appropriate superscript in the formulation kg/m2.
Thank you for your keen observation. We have made corrections to the manuscript regarding the kg/m² superscript.
- Did authors include apyrase or prostaglanding E2 do whole blood before centrifugation to prevent platelets agreggation during PRP obtaining?
Thank you for your excellent comment. Sodium citrate was used as an anticoagulant during the process to prevent the blood from clotting. To further clarify the methodology, we have added this information to the methods section.
- Why authors did not prepare PRP without contamination of WBC? If their PRP contained WBC, they should not named it as PRP.
Thank you for your valuable comment. The method of PRP extraction and manufacturing was described according to GloFinn Corporation’s kit instructions. The final PRP product adhered to the manufacturer's specifications, and the kit instructions were meticulously followed. The manufacturer states that the white blood cell ratio doubles compared to whole blood after centrifugation. However, due to the retrospective nature of this study and the fact that patients had already received the injections, no analyses of the final product could be conducted. Therefore, we lack data concerning the precise ratios of leukocytes. As described in the materials and methods section, the final product was leukocyte-rich platelet-rich plasma (LR-PRP). LR-PRP is defined as having a neutrophil concentration above baseline, and since the leukocyte ratio doubles according to the manufacturer, the neutrophil levels increase above baseline in the final product, thus qualifying it as LR-PRP. We have sharpened this clarification to be as clear as possible in the materials and methods section.
- Did authors verify the platelet count and leukocyte count in obtained PRP? This might be important considering that authors claimed that: "The final PRP contained white blood cells making it leukocyte-rich PRP". What is the percentage of platelets and leukocytes in PRP? There might be some inter-individual differences in count of platelets and leukocytes that could affect the preparation of injected smaples of PRP. If authors measured platelet and leukocyte count, they should inlcude these results in paper.
Thank you for your excellent comment. The study was retrospective, and patients had already received the injections before data collection was performed; therefore, we lack data concerning the precise ratio of leukocytes or platelets. The final product was not void of leukocytes, and since the white blood cell count increases above the whole blood baseline according to the manufacturer, by definition, the final product qualifies as leukocyte-rich PRP (LR-PRP). LR-PRP is defined as having a neutrophil concentration above baseline. The manufacturer reports that the platelet concentration is at least four times above physiological levels. While individual variability in whole blood platelet count may introduce variance to the final PRP platelet count, since all patients with any hematological conditions were excluded, it is safe to assume that none of the patients had platelet counts below the normal baseline in their whole blood, thus reducing the potential variance to merely normal physiological levels. This clarification has been added to the materials and methods section and to the discussion section regarding the strengths and limitations of this study.
- Presentation of results on the Figure 1 is unacceptable. Authos should use a more professional graphical way of presentation their results.
Thank you for your constructive and valuable comment. We have removed the old Figure 1 and created a completely new one to better represent the results with higher resolution.
- Considering the large standard deviation range it is surprising that authors show any significant differences on the Figure 1.
Thank you for your comment. The statistical analyses were described in the materials and methods section, and the results section presented the outcomes of these analyses. Table 2 displays the WOMAC overall scores at the last follow-up for both groups, along with the corresponding p-value from the comparison (p = 0.023), indicating statistical significance. Notably, this follow-up point is where the mean values exhibit the most notable difference between the groups. However, when considering the study's overall results, the discrepancies between the groups are minimal, suggesting that from a clinical perspective, there may be little or no difference in the efficacy of PRP injections for mild to moderate knee osteoarthritis between obese and non-obese patients.
One interpretation of these findings is that obese patients with knee osteoarthritis may derive benefits from PRP treatment in a manner almost identical to non-obese patients. This suggests that such patients could undergo a relatively low-risk treatment, potentially buying time for implementing lifestyle interventions. These interventions could help delay or mitigate the need for arthroplasty, or at the very least, reduce the risks associated with arthroplasty in obese patients. To provide clarity and summarize the interpretation of these results, we have added a section to the final conclusions.
Round 2
Reviewer 2 Report
Comments and Suggestions for Authors
I do not have further comments. Authors responded to all my previous comments.